# Improving the Universality and Learnability of Neural Programmer-Interpreters with Combinator Abstraction

**Da Xiao**[1,2]**, Jo-Yu Liao**[2]**, Xingyuan Yuan**[2]
[1]School of Cyberspace Security, Beijing University of Posts and Telecommunications, China
[2]ColorfulClouds Technology Co., Ltd, Beijing, China
`xiaoda99@gmail.com, {liaoruoyu,yuan}@caiyunapp.com`

## Abstract

To overcome the limitations of Neural Programmer-Interpreters (NPI) in its universality and learnability, we propose the incorporation of combinator abstraction into neural programing and a new NPI architecture to support this abstraction, which we call Combinatory Neural Programmer-Interpreter (CNPI). Combinator abstraction dramatically reduces the number and complexity of programs that need to be interpreted by the core controller of CNPI, while still allowing the CNPI to represent and interpret arbitrary complex programs by the collaboration of the core with the other components. We propose a small set of four combinators to capture the most pervasive programming patterns. Due to the finiteness and simplicity of this combinator set and the offloading of some burden of interpretation from the core, we are able construct a CNPI that is universal with respect to the set of all combinatorizable programs, which is adequate for solving most algorithmic tasks. Moreover, besides supervised training on execution traces, CNPI can be trained by policy gradient reinforcement learning with appropriately designed curricula.

## 1 Introduction

Teaching machines to learn programs is a challenging task. Numerous models have been proposed for learning programs, e.g. Neural Turing Machine (Graves et al., 2014), Differentiable Neural Computer (Graves et al., 2016), Neural GPU (Kaiser & Sutskever, 2015), Neural Programmer (Neelakantan et al., 2015), Neural Random Access Machine (Kurach et al., 2015) and Neural Programmer-Interpreter (Reed & de Freitas, 2016). These models are usually equipped with some form of memory components with differentiable access. Most of these models are trained on program input-output pairs and the neural network effectively learns to become the particular target program, mimicking a particular Turing machine.

Of these models one notable exception is Neural Programmer-Interpreters (NPI) (Reed & de Freitas, 2016) and its extension that supports recursion (Cai et al., 2017) (referred to in this paper as RNPI). NPI has three components: a core controller that is typically implemented by a recurrent neural network, a program memory that stores embeddings of learned programs, and domain-specific encoders that enable a single NPI to operate in diverse environments. Instead of learning any particular program, the core module learns to *interpret* arbitrary programs represented as program embeddings, mimicking a universal Turing machine. This integration of the core (interpreter) and a learned program memory (programmer) offers NPIs with better flexibility and composability by allowing the model to learn new programs by combining subprograms. Despite these merits, the NPI model bears some theoretical and practical limitations that hinder its application in real world problems.

One hypothetical theoretical property of the NPI model that makes it appealing for multi-task, transfer, and life-long learning settings is its *universality*, i.e. the capability to represent and interpret *any* program. As the NPI relies solely on the core to interpret programs, universality requires a fixed core to interpret potentially many programs. A universal fixed core is critical for learning and re-using learned programs in a continual manner, because a core with changing weights may fail to interpret old learned programs after learning new ones. Although the original NPI paper shows empirically

that a single shared core can interpret 21 programs to solve five tasks, and that a trained NPI with fixed core can learn a new simple program MAX, it is unclear whether a universal NPI exists or how universal it could be. Specifically, given the infinite set of all possible programs, the subset of programs that can be interpreted by a fixed core is not explicitly defined. Even though a universal NPI exists, it may still be intractable to provable guarantee of universality by the verification method proposed in Cai et al. (2017), because there may be infinite programs to verify.

Practically, as proposed in Reed & de Freitas (2016), the training of an NPI model relies on a very strong form of supervision, i.e. the example execution traces of programs. This form of training data is typically more costly to obtain than input-output examples. Training with weaker form of supervision is desirable to unlock NPI's full potential.

In this paper, we propose to overcome these limitations of NPI by incorporating combinator abstraction into the NPI model and augmenting the original NPI architecture with necessary components and mechanisms to support this abstraction. We refer to this new architecture as the Combinatory Neural Programmer-Interpreter (CNPI). As an important abstraction technique in functional programming, combinators, a.k.a. higher-order functions, are used to express some common programming patterns shared across different programs. We find that combinator abstraction can dramatically reduce the number and complexity of programs (i.e. combinators) that need to be interpreted by the core, while still allowing the CNPI to represent and interpret arbitrary complex programs by the collaboration of the core with the other components. We propose a small set of four combinators to capture four most pervasive programming patterns. Due to the finiteness and simplicity of this combinator set and the offloading of some burden of interpretation from the core, we are able construct a CNPI with a fixed core that can represent and interpret an infinite number of programs which is adequate for solving most algorithmic tasks. This CNPI is universal with respect to the set of all combinatorizable programs. Moreover, we show empirically that besides supervised training on execution traces, it is possible to train the CNPI by policy gradient reinforcement learning with appropriately designed curricula.

## 2 OVERVIEW OF COMBINATOR ABSTRACTION

### 2.1 REVIEW OF NPI WITH ITS LIMITATIONS

In this section, we give a brief review of the NPI architecture from Reed & de Freitas (2016) and Cai et al. (2017). Then we analyze its limitations to motivate our combinator abstraction. The NPI model has three learnable components: a task-agnostic core controller, a program memory, and domain-specific encoders that allow the NPI to operate in diverse environments. The core controller is a long short-term memory (LSTM) network (Hochreiter & Schmidhuber, 1997) that acts as a router between programs. At each time step, the core can decide either to select another programs to call with certain arguments, or to end the current program. When the program returns, control is returned to the caller by popping the callers LSTM hidden units and program embedding off of a program call stack and resuming execution in this context.

NPI's inference procedure is as follows (see Section 3.1 and Algorithm 1 in Reed & de Freitas (2016) for more detail). At time step $t$, an encoder $f_{enc}$ takes in the environment observation $e_t$ and arguments $a_t$ and generates a state $s_t$. The core LSTM $f_{lstm}$ takes in the state $s_t$, a program embedding $p_t \in \mathbb{R}^P$ and the previous hidden state $h_{t-1}$ to update its hidden state $h_t$. From the top LSTM hidden state several decoders generate the following outputs: the return probability $r_t$, the next program's key embedding $k_{t+1}$, and the arguments to the next program $a_{t+1}$. The next program's ID is obtained by comparing the key embedding $k_{t+1}$ to each row of key memory $M_{key}$. Then the program embedding is retrieved from program memory $M_{prog}$ holding $N$ programs as: $i^* = \arg\max_{j=1..N} (M_j^{key})^T k_{t+1}$ , $p_{t+1} = M_{i^*}^{prog}$.

The above-described NPI architecture bears two limitations. First, as shown in the above equation, at each time step, a decision must be made by the core to select the next program to call out of *all* $N$ currently learned programs in the program memory. As $N$ grows large, e.g. to hundreds or thousands, interpreting programs correctly becomes a more and more difficult task for a single core. What makes things worse is that the core has to learn to interpret new programs without forgetting old ones. Second, it is common that programs with different names and functionalities share some common underlying programming patterns. We take two programs used in Reed & de Freitas (2016)

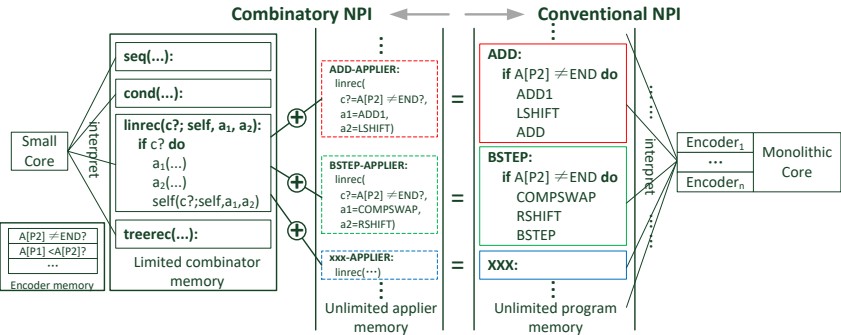

Figure 1: NPI with combinator abstraction.

and Cai et al. (2017) as example (see Figure 1): the ADD1 program in grade school addition, and the BSTEP program to perform one pass of bubble sort. We use their recursive forms described in Cai et al. (2017). The two programs share a very common looping pattern. However, the core needs to learn each of these programs separately without taking any advantage of their shared patterns. The total number of programs that need to be learned by the core thus become infinite. We argue that these two limitations make it very challenging, if not impossible, to construct a universal NPI.

## 2.2 OUR APPROACH USING COMBINATOR ABSTRACTION

To overcome the limitations of the NPI, we propose to incorporate combinator abstraction into the NPI architecture. In functional programming, combinators are a special kind of higher-order functions that serve as power abstraction mechanisms, increasing the expressive power of programming languages. We adapt the concept to neural programming and make it play the central role in improving the universality of NPI.

Conceptually, a *combinator* is a "program template" with blanks as *formal* arguments that are callable as subprograms. An actual program can be formed by wrapping a combinator with another program called an *applier*, which invokes the combinator and passes the *actual* arguments to be called when executing the combinator. Alternatively, an applier applies a combinator to a set of actual programs as callable arguments. Note that the callable arguments themselves can also be wrapped programs (i.e. appliers), and programs with increasing complexity can thus be built up. As in the original NPI, the interpretation of a combinator is conditioned on the output of a lightweight domain-specific encoder which we call an *detector*. It is also provided on the fly by the applier. Figure 1 illustrates the usage of combinator abstraction in the NPI architecture.

In the CNPI architecture, combinators are the only type of programs that need to be interpreted by the core. By prohibiting a combinator to call programs other than those passed to it as arguments, the selection range for next program to call at each time step is reduced from a growing $N$ to a constant $K$, which is the maximum number of arguments for combinators ($\leq 9$ in our proposed model). Meanwhile, compared to the infinity of all possible programs, the number of useful combinators is finite and typically small. In practice, we construct a small set of four combinators to express four most pervasive programming patterns. Therefore, the core only needs to interpret a small number of simple programs. We will show that a quite small core suffices for this job, and that by the collaboration of this core and the other components a universal CNPI can be constructed.

## 3 COMBINATORY NPI MODEL

### 3.1 COMBINATORS AND COMBINATORY PROGRAMS

We propose a set of four combinators to express four most pervasive programming patterns for algorithmic tasks: sequential, conditional, linear recursion and tree recursion (i.e. multi-recursion). The pseudo-code for these combinator are shown in Figure 2. Each combinator has four callable arguments $self$, $a1$, $a2$ and $a3$ and one detector argument $c?$. $self$ is a *default* argument referring

to the combinator itself and is used for recursive call. For linrec and treerec, we give more readable aliases to $a1$, $a2$ and $a3$ to hint their typical roles. The detector argument detects some condition (e.g. a pointer P2 reaching the end of array) in the environment and provides signals for the combinator to condition its execution. It outputs $0$ if the condition satisfies, otherwise $1$. For seq a default blind detector is passed, which always outputs $0$. Although not directly callable, detectors can also be viewed as programs "running in background" as perception modules. In this paper, the conditions to detect is often used to name detectors and we append a '?' to their names to differentiate them from callable programs. Like primitive actions (ACTs), we could also define primitive detectors (DETs) for specific tasks. Note that this combinator set is by no means unique or minimal. They take their current forms mainly for ease of use and learning.

```
# sequential pattern    # conditional pattern    # linear recursion pattern          # tree recursion pattern
def seq(c?; self,        def cond(c?; self,       def linrec(divisible?/c?; self,     def treerec(divisible?/c?; self,
    a1, a2, a3):             a1, a2, a3):             do/a1, next/a2, base/a3):           pre/a1, divide/a2, post/a3):
    a1()                    if c?():                 if divisible?():                    if divisible?():
    a2()                        a1()                     do()                                pre()
    a3()                        a2()                     next()                              _push_sentinel()
                            else:                        self(divisible?; self,              divide()
                                a3()                         do, next, base)                 _mapself(divisible?; self,
                                                     else:                                           pre, divide, post)
                                                         base()                              post()
```

Figure 2: Pseudo-code for the set of combinators.

The four combinators are classified into two categories. seq, cond and linrec are basic combinators, which only call their callable arguments during execution. treerec is an advanced combinator. Besides callable arguments, an advanced combinator can also call built-in programs, such as _push_sentinel and _mapself in treerec. These built-in programs are used to facilitate multiple recursive calls to $self$ in treerec combinator. Basically, $divide$ prepares states necessary for each recursive call and push these states to a stack. The built-in combinator _mapself shares a similar structure with linrec. It loads the states one by one from the stack and makes the recursive call with each state until a sentinel is met (The sentinel is pushed to the stack before $divide$ by _push_sentinel, which is a built-in ACT). More details on built-in programs and treerec are given in Appendix A, and examples of using them can be found in Appendix B.

We now describe how to compose combinator programs using combinators by taking the BSTEP program (i.e. one pass of bubble sort) as example. The normal and combinatory version of the program are shown in Figure 3 (a) and (b) respectively. Recall that an applier applies a combinator to a set of actual programs (ACTs or other predefined appliers) to form a new actual program. Composing a combinatory program amounts to defining appliers iteratively. As shown in Figure 3 (c), during the execution of a combinatory programs, combinators and appliers call each other to form an alternating call sequence until reaching a ACT. Combinators, appliers and detectors are all highly constrained programs, and thus are all easily interpretable and learnable. Nevertheless, they can collaborate to build arbitrarily complex programs.

```
def COMPSWAP:          def COMPSWAP:              BSTEP
  if A[P1]>A[P2]:        cond(A[P1]>A[P2]?;         linrec(..)
    SWAP_12                 SWAP_12, NOP, NOP)        a1->COMPSWAP
def RSHIFT:            def RSHIFT:                     cond(..)
  P1_RIGHT              seq(; P1_RIGHT, P2_RIGHT)       a1->SWAP_12
  P2_RIGHT             def BSTEP:                    a2->RSHIFT
def BSTEP:             linrec(A[P2]≠END?;             seq(..)
  if A[P2]≠END:           COMPSWAP, RSHIFT, NOP)       a1->P1_RIGHT
    COMPSWAP                                          a2->P2_LEFT
    RSHIFT                                        self->linrec(..)
    BSTEP                                            a1->COMPSWAP
                                                  ...
```

| (a) Normal program. | (b) Combinatory program. | (c) Trace of combinatory program. | (d) Interaction between programs and detectors. |

Figure 3: Example combinatory program of BSTEP. NOP is special ACT which does nothing.

## 3.2 CNPI ARCHITECTURE AND ALGORITHM

Having introduced combinators and how to use them to compose combinatory programs, we now describe how these programs are interpreted by the CNPI and the necessary augmentations to the original NPI architecture to enable the interpretation. The complete inference procedure for CNPI is given in Algorithm 1. An example execution of the BSTEP program is illustrated in Figure 4.

Appliers are effectively one-line programs that apply a program $prog$, which could be either a combinator or an ACT, to a set of arguments. To interpret an applier $appl$ we just need to identify the program to be called and its arguments, prepare environment for the invocation, and make the invocation. For easy of interpretation, we propose to store the key embeddings of $prog$ and its detector and callable arguments $c?$, $a1$, $a2$ and $a3$ directly in the applier's program embedding $p_{appl}$:

$$p_{appl} = k_{prog}|k_{c?}|k_{a1}|k_{a2}|k_{a3} \tag{1}$$

where $|$ denotes concatenation [1]. We use a *fixed* parser to extract the key embeddings from the applier's embedding. Then the combinator or ACT ID $i$ is computed by comparing the key $k_{prog}$ with each row of memory $M_{key}$ and finding the best match. The callable arguments' IDs are computed similarly. In CNPI architecture, the models for detectors are stored in a detector memory ($W^{key}$, $W^{weight}$) which has the same key-value structure as the program memory. The detector argument ID $i'$ is computed by comparing $k_{c?}$ to each row of $W_{key}$. Note that the core LSTM does not participate in the interpretation of appliers. As the format for storing these key embeddings is predefined, the fixed parser can parse *any* applier's embedding.

We use a dynamically constructed data structure called a *frame* to pass arguments to a combinator. Each frame is a table of $K$ bindings which associate formal callable arguments with their corresponding actual IDs, with $K$ the number of callable arguments for combinators. When calling a combinator, a new frame is created. The IDs of the combinator's callable arguments (including the combinator's ID $i$ as it corresponds to the $self$ argument) are filled into the frame [2]. In practice we do not use a key-value structure for frames. Instead the frame only stores values, i.e. the arguments' IDs in a fixed order of $self$, $a1$, $a2$ and $a3$.

The interpretation of combinators is in general similar to the inference procedure in Algorithm 1 in Reed & de Freitas (2016). Here we highlight several key differences. In the initialization stage, besides retrieving combinator embedding from the program memory, the detector model is also loaded from the detector memory. Then instead of using the combinator embedding as input to the LSTM at every time step, we use it to initialize the LSTM's state, i.e. each layer's hiddens and cells. We find empirically that this parameterization has better efficiency and accuracy for our combinators; see Section 5.1. The second difference is that we binarize the output of detector $f_{det}$ to get a binary condition $c$ before feeding it to the LSTM:

$$c \leftarrow \mathbb{1}(f_{det}(e) \geq \beta) \, , \, h \leftarrow f_{lstm}(c, h) \tag{2}$$

where $\mathbb{1}()$ is an indicator function. This operation effectively decouples the detector from the core LSTM. This enables us to verify the core's behaviors separately without considering any specific detectors, given that the correct condition is provided. This is difficult to achieve in the original NPI architecture where the core is trained jointly with the encoders.

The third and most important difference is on how the next subprogram to call is computed. We use a decoder $f_{prog}$ to compute a score vector $S \in \mathbb{R}^{\mathbb{K}}$ to assign a score for each formal callable argument. The argument with the maximum score is selected and its actual program ID is retrieved from the frame $F$. This ID is used in turn to retrieve the program embedding from the program memory $M^{prog}$ when the next program is executed:

$$z^* = \arg\max_{j=1..K} S_j \, , \, i^* \leftarrow F[z^*] \, , \, p_{t+1} = M^{prog}_{i^*} \tag{3}$$

where $K$ is the maximum number of callable arguments for combinators. We consider this indirection of subprogram embedding retrieval, together with the dynamic binding of formal arguments to actual programs in the frames, to be the key to the superior universality and learnability of CNPI.

---

[1] For the `seq` combinator and ACTs which do not need detector arguments, the blind detector's key embedding is stored. For ACTs with arguments the arguments' values are stored in place of the key embeddings.

[2] If the combinator is treerec, the IDs of built-in programs also need to be appended to the end of the frame in a predefined order. In this case the frame's size is $K + B$, with $B$ the total number of built-in programs.

When calling the subprogram the same detector ID and frame are re-used, which is equivalent to passing the combinator's arguments to all of its subprograms. This facilitates recursion as these arguments are needed by linrec and treerec when calling $self$ (see Figure 2). For other subprogram calls to appliers or ACTs, these arguments are safely ignored.

---

**Algorithm 1** Combinatory neural programming inference

1: **Inputs:** Environment observation $e$, program ID $i$, detector ID $i'$, frame $F$, stop threshold $\alpha$, condition threshold $\beta$, number of arguments (including $self$) $K$ for combinators
2: **function** RUN($i, i', F$)
3:  $r \leftarrow 0$, $p \leftarrow M_i^{prog}$
4:  **if** $p$ is an applier **then**
5:    $i_2, i_2', a_2 \leftarrow$ _PARSE($p$)        ▷ Get the next program to run with its detector and args.
6:    $F_2 \leftarrow$ FRAME($K$), $F_2[1] \leftarrow i_2$     ▷ New an empty frame and fill in $self$ arg.
7:    **for** $j = 1$ to $K - 1$ **do** $F_2[j + 1] \leftarrow a_2[j]$     ▷ Fill in the other args.
8:    RUN($i_2, i_2', F_2$)           ▷ Run subprogram $i_2$ with detector $i_2'$ and frame $F_2$.
9:    FREE($F_2$)               ▷ Free frame $F_2$'s space.
10:  **else if** $p$ is a combinator **then**
11:    $f_{det} \leftarrow W_{i'}^{weight}$, $h \leftarrow p$      ▷ Load detector from detector memory and init LSTM.
12:    **while** $r < \alpha$ **do**
13:      $c \leftarrow \mathbb{1}(f_{det}(e) \geq \beta)$, $h \leftarrow f_{lstm}(c, h)$      ▷ $c$ is a binary condition.
14:      $r \leftarrow f_{end}(h)$, $S \leftarrow f_{prog}(h)$     ▷ $S$ is a $K$-dim score vector.
15:      $z_2 \leftarrow \arg\max_{j=1..K} S_j$     ▷ Decide the argument ID of the next program to run.
16:      $i_2 \leftarrow F[z_2]$         ▷ Retrieve the ID of the next program to run from frame.
17:      RUN($i_2, i', F$)        ▷ Run subprogram $i_2$ with the same detector and frame.
18:  **else**                       ▷ $p$ is a primitive action.
19:    $a \leftarrow F[2 : K]$, $e \leftarrow f_{env}(e, p, a)$     ▷ Unpack args from $F$ and do the action.
20:
21: **function** _PARSE(p)                ▷ Helper function for interpreting appliers.
22:  $k, k', a \leftarrow$ SPLIT($p$)     ▷ Get program, detector and arg keys from applier embedding.
23:  $i \leftarrow \arg\max_{j=1..N}(M_j^{key})^T k$, $i' \leftarrow \arg\max_{j=1..M}(W_j^{key})^T k'$     ▷ Program key to id.
24:  **if** $i$ is not a primitive action **then**
25:    **for** $j = 1$ to $K - 1$ **do** $a[j] \leftarrow \arg\max_{j'=1..N}(M_{j'}^{key})^T a[j]$   ▷ Arg keys to IDs.
26:  **return** $i, i', a$

---

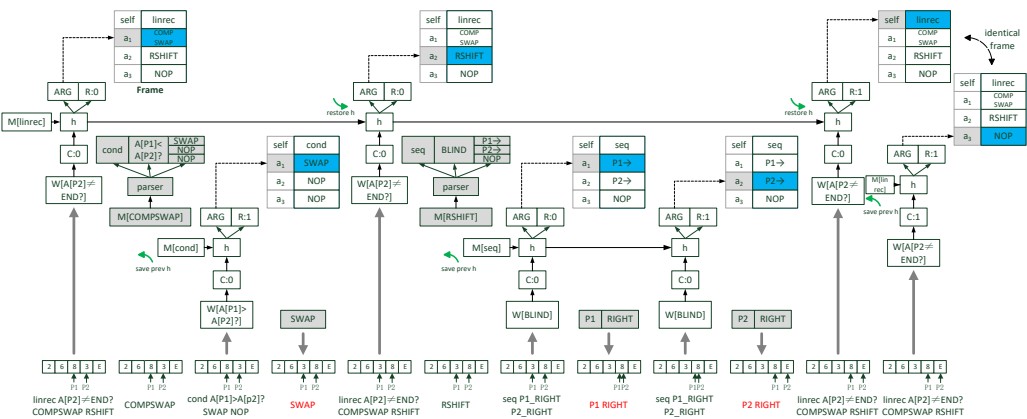

Figure 4: Example execution of the combinator BSTEP program. SWAP, P1_RIGHT and P2_RIGHT are in fact appliers which need to be parsed as COMPSWAP and RSHIFT. We omit this step and treat them as ACTs in the figure for brevity.

## 3.3 TRAINING

CNPI has four components: the core, the program (combinator and applier) memory, the detector memory, and the parser, of which the first three are learnable. The combinators are trained jointly with the core. Detectors and appliers are trained separately.

Supervised learning (SL) of CNPI uses execution traces of combinatory programs. A single element of an execution trace consists of a step input-step out pair, which takes one of the two forms: $\xi_t^{comb}$ : $\{e_t, i_t\} \rightarrow \{c_t, z_{t+1}\}$ for combinator execution and $\xi_t^{appl}$ : $\{i_t\} \rightarrow \{i_{t+1}, i'_{t+1}, a_{t+1}\}$ for applier execution. $z_{t+1}$ is the formal callable argument ID to be called by the applier at time step $t + 1$. $c_t$ is the correct condition at time step $t$ and is used as the output target for detectors. $i_{t+1}$ and $r_t$ provide targets for the core. Detectors and the core are trained on the $\xi_t^{comb}$ elements of the trace, using stochastic gradient ascent to maximize the likelihood of their corresponding targets.

$$\Delta w \propto \sum_{t=1}^{T} (\nabla_w \log p_w(c_t \mid e_t)) \, , \ \Delta\theta \propto \sum_{t=1}^{T} (\nabla_\theta \log p_\theta(i_{t+1} \mid c_t) + \nabla_\theta \log p_\theta(r_t \mid c_t)) \quad (4)$$

where $w$ are parameters of the detector model, $\theta$ are the collective parameters of the core and the combinator embedding, $T$ is the length of the sequence of $\xi_t^{comb}$ elements. The probability $p(i_{t+1} \mid c_t)$ of calling subprogram $i$ is computed by applying a softmax to the scores produced by $f_{prog}$: $p(i_{t+1} \mid c_t) = \exp s(i_{t+1} \mid c_t) / \sum_j \exp s(j_{t+1} \mid c_t)$. In SL the applier embeddings do not need to be trained; they are just generated from $\xi_t^{appl}$ elements of the trace according to equation (1).

CNPI can also be trained by policy gradient reinforcement learning (RL) [3]. No execution trace is given and the core tries to complete the task by making program calls following the probabilities $p(i_{t+1} \mid c_t)$ and feeding-forward the LSTM. An episode ends if the task is completed or the number of steps reaches MAX_NSTEP. In our experiments, MAX_NSTEP = $K \cdot n$, where $K$ is the number of callable arguments for combinators, $n$ is the complexity of the problem to be solved. A reward $R_T$ is given when an episode ends at step $T$:

$$R_T = \begin{cases} +1 - 0.1 \times T & \text{if task is completed} \\ -1 - 0.1 \times T & \text{otherwise} \end{cases} \quad (5)$$

At each time step $t$, a condition $\tilde{c}_t$ is sampled from a Bernoulli distribution defined by the output of the detector. The next program to be called is identified as $F[\tilde{i}_{t+1}]$, where $\tilde{i}_{t+1}$ is sampled from $\{p(i_{t+1} \mid \tilde{c}_t)\}$. The core is trained using stochastic gradient ascent on a mixed objective with two parts: an RL objective of maximizing expected reward, plus an SL objective of maximizing the likelihood of correct flag of program return:

$$\Delta\theta \propto \sum_{t=1}^{T} (\nabla_\theta \log p_\theta(\tilde{i}_{t+1} \mid \tilde{c}_t) R_T + \nabla_\theta \log p_\theta(r_t \mid \tilde{c}_t) \mathbb{1}(R_T > 0)) \quad (6)$$

The RL objective is derived from the REINFORCE algorithm (Williams, 1992). Note that the SL objective only takes effect on episodes where a positive reward is received on task completion. This combination of RL and SL objectives to optimize a policy is also used in Oh et al. (2017) to learn parameterized skills. The detector is also trained to maximize expected reward using REINFORCE:

$$\Delta w \propto \sum_{t=1}^{T} (\nabla_w \log p_w(\tilde{c}_t \mid e_t) R_T) \quad (7)$$

Once the detector and the core have been learned, applier embeddings can also be learned using RL. After the program $\tilde{i}$ is called with detector and callable arguments $\tilde{i}'$ and $\tilde{a}_j, j = 1..K - 1$, a reward $R \in \{-1, +1\}$ is given according to whether the task has been completed. The applier embedding parameterized by $\phi$ is updated as:

$$\Delta\phi \propto \nabla_\phi \left( \log p_\phi(\tilde{i}) + \log p_\phi(\tilde{i}') + \sum_{j=1}^{K} \log p_\phi(\tilde{a}_j) \right) R \quad (8)$$

---

[3]In this paper we only train CNPI by RL on tasks that can be solved by the three basic combinators.

where the identifiers $\tilde{i}$, $\tilde{i}'$ and $\tilde{a}_j$ are sampled respectively from the distributions derived from the corresponding keys stored in the applier's embedding: $\tilde{i} \sim \text{softmax}(M^{key}k_{prog})$, $\tilde{i}' \sim \text{softmax}(W^{weight}k_{c?})$, $\tilde{a}_j \sim \text{softmax}(M^{key}k_{a_j})$, $j = 1..K-1$.

Note that in both SL and RL, detectors are trained separately from the core. This decoupling facilitates the sharing of detectors across programs and the verification of the behavior of the core.

## 4 ANALYSIS

Training CNPI with SL to solve algorithmic tasks consists of three steps. First, train and verify the core jointly with the combinators with synthetic *abstract traces*, i.e. sequences of $\xi_t^{comb}$ elements corresponding to the correct invocation of formal callable arguments given conditions (a total of 11 traces for the four combinators). After having been verified for correct behavior, the core and the combinator embeddings are fixed. This step is done only once before solving any specific task. Second, for a new task, identify the conditions needed to solve the task, train and verify detectors to detect these conditions, and then add them to the detector memory. Finally, iteratively define appliers from the bottom up by adding them to the program memory with program embeddings set according to equation (1) given $\xi_t^{appl}$ elements of the traces, and call the topmost applier to solve the task. We state the universality of CNPI with the following theorem and proposition:

**Theorem 1.** *If 1) the core along with the program embeddings of the set of four combinators and the built-in combinator _mapself are trained and verified before being fixed, and 2) the detectors for a new task are trained and verified, then CNPI can 1) interpret the combinatory programs of the new task correctly with perfect generalization (i.e. with any input complexity) by adding appliers to the program memory, and 2) maintain correct interpretation of already learned programs.*

**Proposition 1.** *Any recursive program is combinatorizable, i.e., can be converted to a combinatory equivalent.*

Theorem 1 states that CNPI is universal with respect to the set of all combinatorizable programs and that appliers can be continually added to the program memory to solve new tasks. Proposition 1 shows that this set of programs is adequate for solving most algorithmic tasks, considering that most, if not all, algorithmic tasks have a recursive solution. We prove Theorem 1 in Appendix C. For Proposition 1, instead of giving a formal proof, we propose a concrete algorithm for combinatorizing any program set expressing an recursive algorithm in Appendix B.

We argue that universality is a property harder to achieve than the generalization property discussed in Cai et al. (2017), which provides provable guarantees of perfect generalization for several programs. However, the authors did not consider the problem of universality with a fixed core. In fact, although RNPI can be trained on a particular task and verified for perfect generalization, after training on a new task causing changes to the parameters of the core, the property of perfect generalization on old tasks may not hold any more. In contrast, CNPI provides both generalization and universality. Table 1 qualitatively compares CNPI with NPI and RNPI.

Table 1: Qualitative comparison of CNPI with NPI and RNPI.

| Model | provable perfect generalization | provable universality | # verifications of programs / combinators | encoders / detectors | # trainings of programs / combinators | encoders / detectors |
|---|---|---|---|---|---|---|
| NPI | × | × | — | — | per task | per task |
| RNPI | ✓ | × | per task | per task | per task | per task |
| **CNPI** | ✓ | ✓ | **once** | **per condition** | **once** | **per condition** |

Due to the decomposition of programs into combinators and appliers, and the decoupling of detectors from the core, we can verify the perfect generalization of a particular program using much fewer test inputs than RNPI. For example, to verify the perfect generalization of bubble sort with RNPI we need 2078 test inputs for 6 subprograms while with CNPI we need only 123 for 4 detectors.

## 5 EXPERIMENTS

While the previous section analyzes the universality of CNPI, this section shows results on the empirical evaluation of its learnability via both SL and RL experiments. We mainly report results on learning the core and the combinators, assuming that detectors for the tasks have been trained. Learning a detector in our CNPI architecture is a standard binary classification problem, which can be trivially solved by training a classifier.

To evaluate how CNPI improves learnability over the original NPI architecture, in some experiments we use the RNPI model as a baseline. It has the same architecture as NPI and allows recursive calls. For a CNPI with $K$ callable arguments (denoted as CNPI-$K$), we construct a counterpart RNPI with $K \cdot n$ existing actual programs (either composite programs or ACTs) in the program memory as base programs (denoted as RNPI-$K$x$n$). These base programs are divided into $n$ sets corresponding to $n$ different tasks (e.g. grade school addition and bubble sort). Then new programs are learned over each set by calling the corresponding $K$ base programs as subprograms. Note that some of these new programs may share same patterns (e.g. ADD1 and BSTEP, we call them isomorphic programs), but in RNPI they are treated as different programs and the core needs to learn all of them. For fair comparison, the counterpart RNPI uses the same detector as the CNPI.

For all experiments, we used a one-layer LSTM for the core. We trained the CNPI using plain SGD with batch size 1, and learning rate of 0.5 and 0.1 for SL and RL experiments respectively. For the SL experiments, the learning rate was decayed by a factor of 0.1 if prediction accuracy did not improve for 10 epochs.

### 5.1 SUPERVISED LEARNING RESULTS

We found that, as expected, a CNPI can be trained to learn the set of four combinators using synthetic abstract traces without any difficulty. From Section 4 we know that this CNPI is able to learn all combinatorizable programs (including the four in Appendix B) with perfect generalization. To further stress the learning capacity of the core, we enlarge the small set of four combinators to a full set of all possible combinators with the following two constraints: 1) branching can only happen at the beginning of the execution; 2) call to the $self$ argument, i.e. recursive call, can only be made at the end of the execution (i.e. only tail recursion is allowed). For $K = 4$, this full set has 57 combinators (including the three basic combinators).

Cores with different number of LSTM cells were trained to learn this full set of combinators. We compare two methods of feeding the combinators embedding to the core LSTM: use the embedding as input to the LSTM at every time step (Emb-as-Input), as is done in Reed & de Freitas (2016), and using it as the initial state (i.e. hiddens and cells) of the LSTM (Emb-as-State0). For both methods the combinator embedding size is set to be equal to the LSTM size. Note that the Emb-as-State0 model has fewer parameters than Emb-as-Input with the same number of cells. We also trained a sequence-to sequence model from Sutskever et al. (2014) where an encoder LSTM takes in the text code representation of the combinator (a simplified version of the pseudo-code in Figure 2) and the last state of the encoder is used as the combinator embedding to initialize the core LSTM's state. This seq2seq model can be seen as a miniature of an instruction-to-action architecture. We see in Figure 5 that the Emb-as-State0 model achieves better prediction accuracy than the Emb-as-Input model with the same size. Particularly, the Emb-as-State0 model with only 5 cells can learn all the 57 combinators with 100%. This LSTM is much smaller than the one used in Reed & de Freitas (2016) which has two layers of size 256. The seq2seq model can also achieve 100% accuracy with 7 cells. In subsequent experiments we used a core LSTM of size 16 if not mentioned otherwise.

We compare the abilities of CNPI and RNPI to learn new combinators/programs with a fixed core. The models were first trained on a combinator/program set to get 100% accuracy, then trained on a new set with the core fixed. Finally the models were tested on the old set to see if they are still remembered by the models. For the CNPI-4 model the old and new combinators were generated by a random even split of the full combinator set. For the RNPI-4x2 model with two sets of base programs, we constructed a full set of all possible composite programs for each set of base programs, as with combinators. Then old and new programs were randomly sampled from the two sets respectively. Note that the old and new programs generated this way have certain proportion of isomorphic programs, and this proportion grows with the percentage of random sampling. In the RNPI

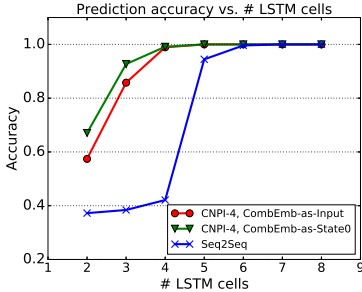

Figure 5: Prediction accuracy on the full combinator set with 4 callable arguments.

| Model | Old / New set | Train old | Train new w/ fixed core | Test old |
|---|---|---|---|---|
| RNPI-4x2 | 50% full set 1 / | 100 | > 90 | 12.9 |
| | 50% full set 2 | 100 | > 97 | 3.3 |
| | 100% full set 1 / | 100 | > 90 | 6.5 |
| | 100% full set 2 | 100 | > 97 | 1.7 |
| **CNPI-4** | 50% full set / the other 50% | 100 | 97.7 | **100** |

Table 2: % accuracy of learning new programs/combinators and remembering old ones with a fixed core. The maximum accuracy obtained when training on new set are 100% and 97.7% for RNPI and CNPI respectively.

experiment, the program key embeddings need to be learned jointly with the program embeddings, otherwise the model would not be able to learn the new programs. As shown in Table 2, although both models can be trained on the new set with high accuracy, when tested on old ones, RNPI shows catastrophic forgetting, which becomes more severe as there are more isomorphic programs between the old and new set. In contrast, CNPI remembers old combinators perfectly.

## 5.2 Reinforcement learning results

We find that curriculum learning is necessary for training CNPI with RL. Table 3 shows the curriculum we used for training CNPI for the sorting task. For each subtask, the programs to be learned (including detectors) are bolded and colored. The curriculum has two stages. In the first stage, the combinators were trained with simple auxiliary tasks, using ACTs and DETs as arguments. The learned combinators are then used as prerequisite for solving the actual tasks. The tasks for each combinator is designed to ensure that the task will be completed if and only if the combinator is correctly executed as defined in Figure 2. In the second stage, detectors and appliers can be learned in two forms: we can either define a sketch for solving the task (similar to the policy sketches in Andreas et al. (2017)), with some learnable arguments (as in the Compare and swap and Output max tasks), or learn an applier to solve the task using already learned arguments (as in the Sort task). Note that for brevity we define some appliers for resetting pointers directly without any learning after at the end of Stage 1. In fact, they could also be learned the same way as the appliers in Stage 2.

Though being quite simple programs, we find that the three basic combinators are still difficult to learn with plain policy gradient RL, even with the curriculum. To facilitate learning. we use the adaptive sampling technique proposed in Reed & de Freitas (2016). Example traces are fetched with frequency proportional to the models current prediction error. We set the sampling frequency using a softmax over a moving average of prediction error over last 10 episodes, with temperature 1. Besides, for each combinator's auxiliary task we design an easy version of the task, which corresponds to a partial completion of the true task (see Table 3). Then a curriculum can be formed for each combinator by either mixing the easy and true task (mixed), or complete the easy task first before going to the true one (gradual). For each different use of adaptive sampling and curriculum we ran 100 experiments with a maximum number of 5000 episodes for each experiment. Table 4 shows the success rate that all three auxiliary tasks are completed along with success rates for completing the task for each combinator. As shown in Table 4, both adaptive sampling and the curriculum help training considerably. A relatively high success rate of 91% can be obtained with adaptive sampling and the gradual curriculum. We can also know from Table 4 that of the three combinators seq is the easiest to learn while linrec is the hardest.

We compare the success rate of training CNPI (CNPI-4) with its counterpart RNPI models RNPI-4x$n$. For each combinator's auxiliary task, we constructed $n$ different versions of the task by providing different ACTs and DETs. For example, for the Copy task we used different pointers (e.g. P1) to move in different directions (e.g. to left), and output different symbols when finished (e.g. 'DONE') to generate $3 \times n$ tasks. Then we trained RNPI to learn $3 \times n$ actual programs in parallel to solve these tasks. The RNPI-4x$n$ models were trained with adaptive sampling and gradual curriculum for

Table 3: Curriculum for training CNPI for the sorting task using RL. The programs to be learned (including detectors) are bolded and colored. Several ACTs are added to help learn the tasks: OUT_x: write the element at pointer x to position pointed by P3 and advanced P3 one step. CLEAR_x: set the element at pointer x to $-1$. OUTCLEAR_x: output then clear x. For the sort task, instead of sorting in-place, the max element found in each pass is written to a second array pointed to by P3.

| Subtask | Description | Program to learn |
|---|---|---|
| | Stage 1: Learning the core and combinators | |
| Swap and output easy | Output A[P1] | **seq0**(; OUT_1, NOP, NOP) |
| Swap and output | Output A[P1], then swap A[P1] and A[P2], finally output A[P2] | **seq**(; OUT_1, SWAP_12, OUT_2) |
| Conditional output easy | Output A[P2] if P2 is in array, otherwise output 'OK' | **cond0**(A[P2]≠END?; OUT_2, NOP, OUT_OK) |
| Conditional output | Output and clear A[P2] if P2 is in array, otherwise output 'OK' | **cond**(A[P2]≠END?; OUT_2, CLEAR_2, OUT_OK) |
| Copy easy | Output the first element if array is empty, otherwise output 'OK' | **linrec0**(A[P2]≠END?; OUT_2, NOP, OUT_OK) |
| Copy | Output elements sequentially till the end of array, then output 'OK' | **linrec**(A[P2]≠END?; OUT_2, P2_RIGHT, OUT_OK) |
| Reset pointers | Reset P1 and P2 to the appropriate beginning position | RESET_1: linrec(A[P1]≠END?; P1_LEFT, NOP, NOP) RESET_2: linrec(A[P2]≠END?; P2_LEFT, NOP, P2_RIGHT) RESET: seq(; RESET_1, RESET_2, NOP) |
| | Stage 2: Learning detectors and appliers | |
| Conditional swap | Conditionally swap two elements | COMPSWAP: cond(**A[P1]>A[P2]?**, SWAP_12, NOP, NOP) |
| Output max | Find and output the max element in the array then clear it | MAX: linrec(A[P2]≠END?; **STEP**, P2_RIGHT, OUTCLEAR_1) |
| Sort | Sort the array by repeatedly outputting the current max element | **SORT**: linrec(A[P3]≠END?; MAX, RESET, NOP) |

Table 4: % success of training the core+combinators with RL. The three figures in brackets represent % success of learning the seq, cond or linrec combinator respectively.

| Sampling method | No curriculum | Mixed curriculum | Gradual curriculum |
|---|---|---|---|
| Uniform | **7** (11/10/7) | **17** (33/28/17) | **49** (94/49/49) |
| Adaptive | **31** (74/41/31) | **78** (87/80/78) | **91** (92/91/91) |

Table 5: Comparison of % success of training CNPI with RL with other models.

| Stage | CNPI-4 | RNPI-4x2 | RNPI-4x3 |
|---|---|---|---|
| Easy | **99** | 25 | 0 |
| Final | **91** | 0 | 0 |

$10000 \times n$ episodes, and the success rate over 100 trials are shown in Table 5. Due to the enlarged candidate set of the next program to call from 4 to $4 \times n$ and the increased number of programs to be learned from 3 to $3 \times n$, it is much more difficult to train RNPI with RL. RNPI-4x2 finishes the first stage of the curriculum to complete the easy tasks with a success rate of 25% while fails completely on the final stage to complete the true tasks. RNPI-4x3 can not even finish the easy stage.

We trained the A[P1]>A[P2]? detector in the Conditional swap task with the RL objective. The input to the detector is the one-hot encoding of the two elements. Then the STEP applier was learned in the context of the Output max task by maximizing the expected reward of completing the task; see equation (8). The embedding of STEP was learned successfully in 79% out of the 100 experiments we ran. In each successfully trial one of the two appliers, seq(; COMPSWAP, P1_LEFT, NOP) and cond(A[P1]>A[P2]?; NOP, NOP, MOVE_12) was learned, which was equivalent to finding the max element by a pass of bubble sort and selection sort respectively. MOVE_12 is a primitive applier we defined to move P1 forward until reaching P2. Finally, the Sort applier was learned to complete the Sort task, with success rate of 62% over 100 experiments. Both bubble sort and selection sort were learned by calling the two learned STEP appliers respectively.

## 6  CONCLUSION

The problem of improving the universality and learnability of NPI is addressed for the first time by incorporating combinator abstraction from functional programming. Analysis and experimental results have shown that CNPI is universal with respective to all combinatorizable programs and can be trained with both strong and weak supervision. We believe that the proposed approach is quite general and has potential applications besides solving algorithmic tasks. One scenery is training agents by RL to follow instructions and generalize (e.g., Oh et al. (2017), Andreas et al. (2017), Denil et al. (2017)). Natural language contains "higher-order" words such as "then" and "twice", which play critical role but the interpretation of which may cause trouble to vanilla sequence-to-sequence models (Lake & Baroni, 2017). By representing these words as combinators and equipping the agent with CNPI-like components, it would be possible to construct agents that display more complex and structured behavior and that generalize better. We leave this for future work.

### ACKNOWLEDGMENTS

We thank Mingli Yuan for valuable discussion and feedback.

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

## A  BUILT-IN PROGRAMS TO SUPPORT TREE RECURSION.

We use some built-ins facilities, including a state stack, a combinator, four ACTs and a detector, to support tree recursion. They are listed in Table 6. The pseudo-code for the built-in combinator _map_self is shown in Figure 6. The built-in ACT _load_state need to be overloaded for each specific task because the state needed to for a recursive call may be different for each task. See examples in B for the usage of _load_state.

| Type | Name | Descriptions |
|------|------|--------------|
| data structure | state state | A stack to hold states for recursive calls |
| combinator | _mapself | Make recursive call for each state on stack |
| ACT | _push_sentinel | Push a sentinel to stack to terminate recursive call loop |
| | _push | Push a state to stack |
| | _pop | Pop a state from stack |
| | _load_state | Load the state on top of stack before recursive call |
| detector | _top≠SENTINEL? | Detect the termination condition for recursive call loop |

Table 6: Built-ins to support tree recursion. The state can be seen as arguments for a recursive call.

```
# used by treerec
def _mapself(divisible?/c?; self,
      pre/a1, divide/a2, post/a3):
   if _top≠SENTINEL?():
      _load_state()
      self(divisible?; self,  pre, divide, post)
      _pop()
      _mapself(divisible?; self, pre, divide, post)
   else:
      _pop()   # pop the sentinel
      _load_state()
```

Figure 6: Built-in combinator _mapself used by treerec.

## B  COMBINATORY PROGRAMS FOR ALGORITHMIC TASKS

Below we show the combinatory programs compared with the corresponding normal programs for three algorithmic tasks bubble sort, quick sort and traverse in topological sort. Bubble sort has a nested two levels of linear recursion both of which are expressed by linrec. Quick sort uses bi-recursion and traverse in topological sort uses multi-recursion. Both are expressed by treerec together with SAVE_STATE and _load_state.

Normal and combinatory programs for bubble sort.

```
1   def COMPSWAP():
2     if A[P₁]>A[P₂]:
3       SWAP(P₁,P₂)
4
5   def RSHIFT():
6     MOVE(P₁,UP)
7     MOVE(P₂,UP)
8
9   def BSTEP():
10    if A[P₂] ≠ END:
11      COMPSWAP()
12      RSHIFT()
13      BSTEP()
14
15  def LSHIFT():
16    if A[P₁]≠ END:
17      MOVE(P₁,DOWN)
18      MOVE(P₂,DOWN)
19      LSHIFT()
20
21  def RESET():
22    LSHIFT()
23    MOVE(P₃,UP)
24
25  def BUBBLESORT():
26    if A[P₃]≠END:
27      BSTEP()
28      RESET()
29      BUBBLESORT()
```

(a) Normal Program

```
1   def COMPSWAP:
2     cond(A[P₁]>A[P₂]?; SWAP_12, NOP, NOP)
3
4   def RSHIFT:
5     seq(; P1_RIGHT, P2_RIGHT, NOP)
6
7   def BSTEP:
8     linrec(A[P₂]≠END?; COMPSWAP, RSHIFT,
          NOP)
9
10  def LSHIFT:
11    linrec(A[P₁]≠END?; P1_LEFT, P2_LEFT,
          NOP)
12
13  def RESET:
14    seq(; LSHIFT, P3_RIGHT, NOP)
15
16  def BUBBLESORT:
17    linrec(A[P₃]≠END?; BSTEP, RESET, NOP)
```

(b) Combinatory Program

Normal and combinatory programs for quick sort.

```
1   def COMPSWAP():
2     if A[P_j]≤A[P_hi]:
3       SWAP(P_pivot, P_j)
4       MOVE(P_pivot, UP)
5
6   def COMPSWAP_LOOP():
7     if P_j≠P_hi:
8       COMPSWAP()
9       MOVE(Pj, UP)
10      COMPSWAP_LOOP()
11
12  def PARTITION():
13    SET_PIVOT_LO()
14    SET_J_LO()
15    COMPSWAP_LOOP()
16    SWAP(P_pivot ,P_hi)
17    SET_J_NULL()
18
19  def QUICKSORT():
20    if P_lo<P_hi:
21      PARTITION()
22      STACK(STACK_PUSH_CALL2)
23      STACK(STACK_PUSH_CALL1)
24      WRITE(P_hi, ENV_STACK_HI_PEEK)
25      WRITE(P_lo, ENV_STACK_LO_PEEK)
26      QUICKSORT
27      STACK(STACK_POP)
28      WRITE(P_hi, ENV_STACK_HI_PEEK)
29      WRITE(P_lo, ENV_STACK_LO_PEEK)
30      QUICKSORT
31      STACK(STACK_POP)
```

(a) Normal Program

```
1   def PRE_COMPSWAP_LOOP:
2     seq(; SET_PIVOT_LO, SET_J_LO, NOP)
3
4   def COMPSWAP:
5     cond(A[P_j]≤A[P_hi]?; SWAP_PIVOTJ,
            PPIVOT_RIGHT, NOP)
6
7   def COMPSWAP_LOOP:
8     linrec(P_j≠P_hi?; COMPSWAP, PJ_RIGHT, NOP)
9
10  def POST_COMPSWAP_LOOP:
11    seq(; SWAP_PIVOTHI, SET_J_NULL, NOP)
12
13  def PARTITION:
14    seq(; PRE_COMPSWAP_LOOP, COMPSWAP_LOOP,
            POST_COMPSWAP_LOOP)
15
16  def SAVE_STATE2:
17    _push(P_pivot+1, P_hi)
18
19  def SAVE_STATE1:
20    _push(P_lo, P_pivot-1)
21
22  def DIVIDE:
23    seq(; PARTITION, SAVE_STATE2,
            SAVE_STATE1)
24
25  def _load_state:
26    Write value pair on top of the state
            stack to P_hi and P_lo
27
28  def QUICKSORT:
29    treerec(P_lo<P_hi?; NOP, DIVIDE, NOP)
```

(b) Combinatory Program

Normal and combinatory programs for traverse in topological sort.

```
1   def TRAVERSE():
2     if Q_color(v) is WHITE:
3       WRITE(COLOR_CURR, COLOR_GREY)
4       while Q_color(DAG[v][childList[v]]) is
            valid:
5         WRITE(ACTIVATE_NEIGHB)
6         TRAVERSE()
7         MOVE(ChildList[v], UP)
8       WRITE(COLOR_CURR, COLOR_BLACK)
9       WRITE(RESULT)
```

(a) Normal Program

```
1   def PRE:
2     WRITE(COLOR_CURR, COLOR_GREY)
3
4   def SAVE_STATE:
5     seq(; WRITE(ACTIVATE_NEIGHB), _push(v),
            NOP)
6
7   def DIVIDE:
8     linrec(Q_color(DAG[v][childList[v]]) is
            valid?; SAVE_STATE,
            MOVE(ChildList[v], UP), NOP)
9
10  def _load_state:
11    Write value on top of the state stack
            to v
12
13  def POST:
14    seq(; WRITE(COLOR_CURR, COLOR_BLACK),
            WRITE(RESULT), NOP)
15
16  def TRAVERSE:
17    treerec(Q_color(v) is WHITE?; PRE,
            DIVIDE, POST)
```

(b) Combinatory Program

A general algorithm for converting any program set expressing an recursive algorithm to a combinatory one is given in Algorithm 2. For a program it first removes any multiple recursive calls by using _push_state and _mapself, then removes any loop by replacing them with tail recursion. Finally an iterative maximum matching procedure is used to convert the program to a set of appliers iteratively. We put forward a proposition that any recursive program can be combinatorized in this way. Note that non-recursive programs, (e.g. the stack-based iterative program for topological sort used in Cai et al. (2017)) may still be combinatorized by Algorithm 2, but the process is less straightforward.

---

**Algorithm 2** Convert a recursive program set to a combinatory one.

---

1: **Inputs:** A program set $P$ for solving a task by a recursive algorithm
2: **Outputs:** A combinatory program set $Q$ equivalent to $P$
3: **function** CONVERT($P$)
4:     $Q \leftarrow \{\}$
5:     **for** all subprogram $p \in P$ **do**
6:         Find any multiple recursive calls to the same function (including recursive calls in a loop) and replace them with a _push_state(). If any replacement takes place, add a _push_sentinel() before the first of these calls and add a _mapself() after the last of these calls
7:         Find any loop (without recursive calls in the body) and replace it with a tail recursion
8:         **while** $p$ is not an applier **do**             ▷ Match combinators in descending order of complexity.
9:             MATCHANDREPLACE($p$, treerec, $Q$)
10:             MATCHANDREPLACE($p$, linrec, $Q$)
11:             MATCHANDREPLACE($p$, cond, $Q$)
12:             MATCHANDREPLACE($p$, seq, $Q$)
        **return** $Q$
13:
14: **function** MATCHANDREPLACE($p, comb, Q$)
15:     **for** all block $b \in p$ **do**
16:         **if** $b$ matches the pattern expressed by $comb$ **then**
17:             replace $b$ with an applier $appl$ calling $comb$
18:             $Q \leftarrow Q \cup \{appl\}$

---

## C   PROOF OF THEOREM 1

Before proving Theorem 1, we first give a formal definition of combinatory programs and a lemma on the interpretation of appliers.

**Definition 1**

1. An ACT is a combinatory program.

2. A program with an applier $app$ as entrance is a combinatory program if all of $app$'s callable arguments are combinatory programs.

3. Only that which can be generated by the clause 1-2 in finite steps is a combinatory program.

**Lemma 1.** *If all key embeddings of programs and detectors have unit norm, an applier with program embedding set according to equation (1) is guaranteed to be interpreted correctly, i.e., _Parse in Algorithm 1 outputs correct IDs for the combinator to be called, its detector and callable arguments.*

*Proof.* Because all key embeddings in program key memory $M^{key}$ and detector key memory $W^{key}$ (in this proof we use $M^{key}$ to denote both $M^{key}$ and $W^{key}$ for convenience) have unit norm, the dot product of any two keys equals to their cosine similarity ($S_c$). Suppose a key embedding $k$ in the right-hand side of equation (1), which will be output by Split in line 22 of Algorithm 1, is set as $k = M_i^{key}$, then for any $j \neq i$, $(M_i^{key})^T k = S_c(M_i^{key}, k) = 1 > S_c(M_j^{key}, k) = (M_j^{key})^T k$. According to lines 23-25 of Algorithm 1, the correct program (the combinator, detector or callable arguments) ID, namely $i$, will be selected, guaranteeing the correct interpretation of the applier. $\square$

Note that the unit norm constraint for key embeddings is convenient to satisfy in practice.

Following the above recursive definition of combinatory programs and the procedure of iteratively adding appliers to the program memory from the bottom up, we give an induction proof of Theorem 1. The distinguishing feature of CNPI that enables this proof is the dynamic binding of formal detectors and callable arguments to actual programs, which makes verification of combinator's execution (by the core) and verification of their invocation (by appliers) independent of each other. In contrast, it is impossible to conduct such a proof with NPI and RNPI which lack this feature.

*Proof.* **Base case:** It is obvious that programs composed of a single ACT (including built-in ACT) can be interpreted correctly with perfect generalization (abbreviated as *perfectly interpretable*).

**Induction step:** Assume that the programs referenced by the callable arguments of an applier $app$ are all perfectly interpretable, we prove that program $prog$ with $app$ as entrance is perfectly interpretable. Firstly, from Lemma 1 when $app$ is interpreted the right combinator will be invoked with

the right detector and callable argument IDs. Secondly, because the combinators and the detectors have been verified, the programs referenced by the callable arguments of $app$ are guaranteed to be called at the right time. Finally, when these programs are called, they can be perfectly interpreted. Put it all together, $prog$ can be interpreted correctly. Besides, as the calls to $self$ argument which support recursion are also guaranteed to be made at the right time (in linrec and _mapself), $prog$ can be interpreted correctly with any input complexity, i.e. with perfect generalization.

When adding new detectors/appliers to detector/program memory, the weights of the core, key embeddings and program embeddings of combinators and existing appliers are all hold fixed. Thus, the correct interpretation of learned programs composed of these existing appliers can be proved in exactly the same way, i.e., CNPI maintains correct interpretation of already learned programs. □

