# OpenReview forum: "Improving the Universality and Learnability of Neural Programmer-Interpreters with Combinator Abstraction"
_ICLR.cc/2018/Conference — Accept (Poster)_

### Official Review · AnonReviewer1 · 2017-11-24
**The paper clearly breaks the submission guidelines. The paper is far too long, 14 pages (+refs and appendix, in total 19 pages), while the page limit is 8 pages (+refs and appendix). Therefore, the paper should be rejected.**

**Rating:** 3
**Confidence:** 4

**Review:**

The paper is interesting to read and gives valuable insights.

However, the paper clearly breaks the submission guidelines. The paper is far too long, 14 pages (+refs and appendix, in total 19 pages), while the page limit is 8 pages (+refs and appendix). Therefore, the paper should be rejected. I can not foresee how the authors should be able to squeeze to content into 8 pages. The paper is more suitable for a journal, where page limit is less of an issue.

---

> ### Public Comment · (anonymous) · 2017-12-07
> **Solid contribution and insightful communication should not be simply declined by formatting rules.**
>
> I read through this submission and found very interesting and potentially applicable models in this work. I carefully learnt the above 3 reviewer's comments and agreed that this submission provided a solid contribution in the field of reinforcement learning.
>
> More implications (possible real application scenarios) could be added briefly at last. The article was well organized in an explicit and transparent way, which is good for the community. The length problem could be addressed by moving some technique detailed illustration to the open-source platform to shorten the manuscript.
>
> In all, considering the circulation period, in my opinion, I am very looking forward to discussing with the authors in the conference rather than journals.

---

> > ### Author Response · Authors · 2017-12-15
> > **Thanks very much for your interest in our paper**
> >
> > We have shortened the paper from 14 to 12 pages. We also added a discussion Section in the revision to discuss the potential applications of CNPI in other "more real" domains. Any additional questions or feedback are welcome.

---

> ### Author Response · Authors · 2017-12-15
> **Response to Reviewer 1**
>
> Thanks very much for your comments. The original version is indeed too long. We have uploaded a revision. We have shortened the paper from 14 to 12 pages (+refs and appendix, from 19 to 17 pages) while preserving most of the important contents by:
> 1) abbreviating the description of NPI and removing NPI's inference algorithm (Algorithm 1 in the original version) in Section 2.1;
> 2) rewriting some paragraphs (especially those in Section 5) to make them more succinct;
> 3) substantial re-typesetting (e.g. placing some figures and tables side by side, which is also common practice in other submissions). In order to present the somewhat intricate idea as clear as possible, we use in this paper quite a few figures and tables. The bad typesetting of them in the original version made the manuscript unnecessarily long.
> Considering the dense contents of the paper this is the best that we can do.
>
> We'd like to mention that the 12-page revision is in fact shorter than a number of other submissions, e.g.:
> 1. Modular Continual Learning in a Unified Visual Environment (https://openreview.net/forum?id=rkPLzgZAZ), 14 pages
> 2. Towards Synthesizing Complex Programs From Input-Output Examples (https://openreview.net/forum?id=Skp1ESxRZ), 16 pages
> 3. Sobolev GAN (https://openreview.net/forum?id=SJA7xfb0b), 15 pages
> 4. N2N learning: Network to Network Compression via Policy Gradient Reinforcement Learning (https://openreview.net/forum?id=B1hcZZ-AW), 13 pages

---

> ### Comment · AnonReviewer1 · 2018-01-17
> **The paper presents some very interesting ideas, but the paper is still lengthy and is more suitable for a journal.**
>
> I have now also read the revised version of the paper, i.e., the 12-page version.
>
> The paper is very interesting to read, however slightly hard to digest when you are not familiar with NPI.
>
> The paper presents a clear contribution in addition to previous work, i.e., identifying and proposing a set of four combinators that improves the universality of neural programmer-interpreters. An algorithmic framework is presented for inference and example executions are provided.
>
> The analysis and evaluation of the approach are appropriate, both from a theoretical and experimental point of view. However, the execution examples seem very small and it is hard to predict the generality and scalability of the approach (at least for me). How does the technique scale for large problems / applications?
>
> The paper is very well written and clearly highlights its contribution. However, in my opinion the paper breaks the submission guidelines. The paper is too long, 12 pages (+refs and appendix, in total 17 pages), while the page limit is 8 pages (+refs and appendix). The paper is more suitable for a journal, where page limit is less of an issue.
>
> I have generally a problem with papers that are far too long. The page limits are there for a reason, e.g., all papers should be given an equal amount of space to express the ideas and evaluate them. Although the page limit (8 pages) is a recommendation at this conference, this is the first time I see a paper that breaks / stretches the limit so significantly. I think many of the papers submitted could have been of higher quality, have better evaluations, etc. if they also had stretched the page limits by 50%. I think all papers should be judged based on the same restrictions/limitation, scope, etc.

---

### Official Review · AnonReviewer3 · 2017-11-24
**promising use of functional programming ideas in neural program induction; model description needs clarification**

**Rating:** 7
**Confidence:** 4

**Review:**

Quality
The paper is very interesting and clearly motivated. The idea of importing concepts from functional programming into neural programming looks very promising, helping to address a bit the somewhat naive approach taken so far in the deep learning community towards program induction. However, I found the model description difficult to fully understand and have significant unresolved questions - especially *why* exactly the model should be expected to have better universality compared to NPI and RNPI, given than applier memory is unbounded just like NPI/RNPI program memories are unbounded.

Clarity
The paper does a good job of summarizing NPI and motivating the universality property of the core module.

I had a lot of questions while reading:

What is the purpose of detectors? It is not clear what is being detected. From the context it seems to be encoding observations from the environment, which can vary according to the task and change during program execution. The detector memory is also confusing. In the original NPI, it is assumed that the caller knows which encoder is needed for each program. In CNPI, is this part learned or more general in some way?

Appliers - is it the case that *every* program apart from the four combinators must be written as an applier? For example ADD1, BSTEP, BUBBLESORT, etc all must be implemented as an applier, and programs that cannot be implemented as appliers are not expressible by CNPI?

Memory - combinator memory looks like a 4-way softmax over the four combinators, right? The previous NPI program memory is analogous then to the applier memory.

Eqn 3 - binarizing the detector output introduces a non-differentiable operation. How is the detector then trained e.g. from execution traces? Later I see that there is a notion of a “correct condition” for the detector to regress on, which makes me confused again about what exactly the output of a detector means.

Computing the next subprogram - since the size of applier memory is unbounded, the core still needs to be aware of an unlimited number of subprograms. I must be missing something here - how does the proposed model therefore achieve better universality than the original NPI and RNPI models?

Analysis - for the claim of perfect generalization, I think this will not generally hold true for perceptual inputs. Will the proposed model only be useful in discrete domains for algorithmic tasks, or could it be more broadly applicable, e.g. to robotics tasks?

Originality
This methods proposed in this paper are quite novel and start to bridge an important gap between neural program induction and functional programming, by importing the concept of combinator abstraction into NPI.

Significance
The paper will be significant to people interested in NPI-related models and neural program induction generally, but on the other hand, there is currently not yet a “killer application” to this line of work.

The experiments appear to show significant new capabilities of CNPI compared to NPI and RNPI in terms of better generalization and universality, as well as being trainable by reinforcement learning.

Pros
- Learns new programs without catastrophic forgetting in the NPI core, in particular where previous NPI models fail.
- Detector training is decoupled from core and memory training, so that perfect generalization does not have to be re-verified after learning new behaviors.

Cons
- So far lacking useful applications in the real world. Could the techniques in this paper help in robotics extensions to NPI? (see e.g. https://arxiv.org/abs/1710.01813)
- Adds a significant amount of further structure into the NPI framework, which could potentially make broader applications more complex to implement. Do the proposed modifications reduce generality in any way?

---

> ### Author Response · Authors · 2017-12-15
> **Response to Reviewer 3, Part 1**
>
> Thanks for your very constructive feedback. We have uploaded a revision to incorporate your suggestions. We will try to answer your questions and concerns one by one below.
> > especially *why* exactly the model should be expected to have better universality compared to NPI and RNPI, given than applier memory is unbounded just like NPI/RNPI program memories are unbounded. Also related to:
> > Computing the next subprogram - since the size of applier memory is unbounded, the core still needs to be aware of an unlimited number of subprograms. I must be missing something here - how does the proposed model therefore achieve better universality than the original NPI and RNPI models?
> Re: Applier memory is indeed unbounded. However, the core is in fact *not* aware of any actual applier programs. Let's take the BSTEP program in Figure 4 as an example (also see Figure 3 (c) and line 5-7 and 15-16 of Algorithm 1 in the revision). At the first execution step, the core does not directly call 'COMPSWAP' as the next subprogram. It calls 'a1'. Then the actual subprogram COMPSWAP's ID is looked up in the frame, which is constructed on the fly by the BSTEP applier when calling linrec. The _Parse function in Algorithm 1 and Lemma 1 in Appendix C guarantee that the frame will be filled with correct values.
> In CNPI, the core is only responsible for interpreting combinators and is only aware of formal callable arguments. We offload the responsibility of interpreting appliers from the core to a parser. The two key facts are: 1) the execution of all appliers follows exactly the same pattern: call a combinator with a detector arguments and a fixed number of callable arguments and then return, and 2) the parser itself is a *fixed* program (see function _Parse in Algorithm 1) with no learning taking place at all. As a result, the parser can correctly interpret *any* applier with appropriately set program embeddings (according to equation (1)) regardless of how many applier programs are already stored in the program memory. We propose and prove a lemma (Lemma 1 in Appendix C) on the interpretation of appliers in the revision.
> The distinguishing feature of CNPI that enables this separation of responsibility and that eventually provides the universality of CNPI is the dynamic binding of formal detectors and callable arguments to actual programs. We have rewritten the first half of Section 4 to explicitly propose a theorem and a proposition on the universality of CNPI and added Appendix C in the revision to prove the theorem. Please see the last part of our reply to Review 3's comments for more details.
>
> > Appliers - is it the case that *every* program apart from the four combinators must be written as an applier? For example ADD1, BSTEP, BUBBLESORT, etc all must be implemented as an applier, and programs that cannot be implemented as appliers are not expressible by CNPI?
> Re: Yes. Actually we have proposed a "combinatory programing language" for CNPI where programs are composed by iteratively defining appliers from the bottom up. We give a formal definition of combinatory programs in Appendix C in the revision. We propose a proposition in Section 4 stating that any recursive program is combinatorizable, i.e., can be converted to a combinatory equivalent.
> This proposition shows that the set of all combinatory programs is adequate for solving most algorithmic tasks, considering that most, if not all, algorithmic tasks have a recursive solution. Instead of giving a formal proof of it, we propose a concrete algorithm for combinatorizing any program set expressing an recursive algorithm in Appendix B. Although we believe that the proposition is true (effectively, it says that the combinatory programming language is Turing-complete), we think that a formal proof of it would be too tedious to be included in this paper. The intuition behind is that during the execution of a combinatory programs, combinators and appliers call each other to form an alternating call sequence until reaching a ACT. Arbitrarily complex program structures can be expressed in this way (see the last paragraph of Section 3.1 and Figure 3 (c) and (d)). We'd like to point out that the circle formed by the mutual invocation of combinators and appliers is a very fundamental construct in the interpretation of functional program languages. It can be seen as a "neural equivalent" of the eval-apply circle that lies at the heart of a LISP evaluator. The book "Structure and Interpretation of Computer Programs (2nd edition)" has a good discussion on this (Section 4.1.1, Figure 4.1: The eval-apply cycle exposes the essence of a computer language. https://mitpress.mit.edu/sicp/full-text/book/book-Z-H-26.html#%_sec_4.1.1). The expressive power (Turing-completeness) of functional programming languages like LISP has been well recognized. Anyway, we admit that this is a weakness regarding the theoretical rigor of this paper, which could be improved by future work.

---

> ### Author Response · Authors · 2017-12-15
> **Response to Reviewer 3, Part 2**
>
> > What is the purpose of detectors? It is not clear what is being detected. From the context it seems to be encoding observations from the environment, which can vary according to the task and change during program execution. The detector memory is also confusing. In the original NPI, it is assumed that the caller knows which encoder is needed for each program. In CNPI, is this part learned or more general in some way? Also related to:
> > Eqn 3 - binarizing the detector output introduces a non-differentiable operation. How is the detector then trained e.g. from execution traces? Later I see that there is a notion of a “correct condition” for the detector to regress on, which makes me confused again about what exactly the output of a detector means.
> Re: Your understanding of detectors is basically correct. As described in Section 3.1, the detector, as a lightweight and more "specialized" version of the encoder in NPI, detects some condition (e.g. a pointer P2 reaching the end of array) in the environment and provides signals for the combinator to condition its execution. It outputs 0 if the condition satisfies, otherwise 1. As with the confusion about detector memory, you mentioned "In the original NPI, it is assumed that the caller knows which encoder is needed for each program." This way of saying is not very precise. In the original NPI paper, encoders are constructed and used on a per task basis, rather than per program. All programs of a task use the same encoder, which is predetermined. Once a particular task has been given, the single shared encoder integrates tightly with the core and effectively becomes part of the monolithic model, not subject to any dynamic selection by the core. So no such thing as an encoder memory is needed in NPI. On the other hand, in CNPI it is not until the interpretation of an applier that which detector is needed for the next combinator to call is determined. This detector is then loaded from the detector memory and "attached" to the core. Different programs for the same task may use different detectors (e.g. COMPSWAP and BSTEP for bubble sort task in Figure 3). This architecture is more flexible and promotes the reusability of detectors *across* tasks. For example, a detector detecting pointer P2 reaching the end of array can be used in both grade-school addition task by ADD program and bubble sort task by BSTEP program (see Figure 1). This level of reusability is not easy to achieve in NPI. Reusable detectors can be continually added to the detector memory, just as appliers are added to the program memory, during the lifelong learning process of CNPI to enhance its capability.
>
> > Memory - combinator memory looks like a 4-way softmax over the four combinators, right? The previous NPI program memory is analogous then to the applier memory.
> Re: Whether to use two separate memories for combinators and appliers respectively or to use a single program memory is an implementation issue. While both approaches are feasible, in our current implementation we choose to use a single program memory to store both combinators and appliers and use a flag in the program embeddings to differentiate the two types. Considering this, Figure 1 is a little bit misleading. Anyway, this figure is only for illustration purpose.
>
> > Adds a significant amount of further structure into the NPI framework, which could potentially make broader applications more complex to implement. Do the proposed modifications reduce generality in any way?
> Re: CNPI does add a few essential structures (mainly the frames and the detector memory) into the NPI framework and make the model more complex. But as both the analytical and the experimental results show, the proposed modifications significantly *increase* universality and generality for algorithmic tasks. The limitation is perhaps that we do not discuss how to deal with perceptual input, for which the binary output of detectors may be not sufficient. More detectors types may be needed to extend CNPI to support perceptual inputs, but the proposed detector memory architecture and the dynamic selection of detectors can still be used as basis for such extensions. Overall, we believe that the gains are worth the added complexity.

---

> ### Author Response · Authors · 2017-12-15
> **Response to Reviewer 3, Part 3**
>
> > Analysis - for the claim of perfect generalization, I think this will not generally hold true for perceptual inputs. Will the proposed model only be useful in discrete domains for algorithmic tasks, or could it be more broadly applicable, e.g. to robotics tasks? Also related to:
> > So far lacking useful applications in the real world. Could the techniques in this paper help in robotics extensions to NPI?
> Re: We are not very familiar with robotics, but CNPI does has the potential capability of augmenting intelligent agents trained by RL to follow instructions and do tasks, which may have applications in robotics domain. We discuss this below. Though in this paper we only demonstrate the capability of CNPI in algorithm domain, we believe that the proposed approach is quite general and can potentially be applied to other domains. One such domains is the recent work of treating natural language understanding as inferring and executing programs, applied to semantic parsing for QA (e.g., Andreas et al. (2016), Liang et al. (2017)) and training agents by RL to follow instructions and generalize (e.g., Oh et al. (2017), Andreas et al. (2017), Denil et al. (2017)). Besides normal nouns and verbs, natural language contains ``higher-order'' words such as ``then'', ``if'' and ``until'', which play the critical role of controlling the ``execution'' of other verbs, substantially enhancing the expressive power of the language. Very recently,  Anonymous (2018) shows empirically that the prevalent sequence-to-sequence models struggle at mapping instructions containing these words (e.g., ``twice'') to correct action sequence with good generalization. On the other hand, these words can readily be represented as combinators (e.g., def twice(a): a(); a()). By adding these words to the vocabulary and equipping the agent with CNPI-like components to interpret them as combinators, it would be possible to construct agents that display more complex and structured behavior following succinct instructions, and that generalize better due to the raised level of abstraction. We leave this for future work. We have replaced the conclusion section with a discussion section in the revision to discuss the potential applications of CNPI.
> --------
> Jacob Andreas, Marcus Rohrbach, Trevor Darrell, and Dan Klein. Learning to compose neural networks for question answering. In NAACL, 2016.
> Chen Liang, Jonathan Berant, Quoc Le, Kenneth D. Forbus, and Ni Lao. Neural symbolic machines: Learning semantic parsers on freebase with weak supervision. In Annual Meeting of the Association for Computational Linguistics (ACL), 2017.
> Junhyuk Oh, Singh Satinder, Lee Honglak, and Kholi Pushmeet. Zero-shot task generalization with multi-task deep reinforcement learning. In International Conference on Machine Learning (ICML), 2017.
> Jacob Andreas, Dan Klein, and Sergey Levine. Modular multitask reinforcement learning with policy sketches. In International Conference on Machine Learning (ICML), 2017.
> Misha Denil, Sergio Gómez Colmenarejo, Serkan Cabi, David Saxton, and Nando de Freitas. Programmable agents. arXiv preprint arXiv:1706.06383, 2017.
>
> We hope that these replies and the revision resolve your questions. Any additional questions and suggestions are welcome and we will try our best to make things as clear as possible.

---

### Official Review · AnonReviewer2 · 2017-11-27
**Good Paper**

**Rating:** 7
**Confidence:** 4

**Review:**

The authors propose a variant of the neural programmer-interpreter that can support so called combinators for composing an d structuring computations. In a sense, programs in this variant are at a higher level than those in the original neural programmer-interpreter. The distinguishing aspect of the neural programmer-interpreter is that it learns a generic core (which in the variant of the paper corresponds to an interpreter of the programming language) and programs for concrete tasks simultaneously. Increasing the expressivity of the language with combinators has a danger of making the training of core very difficult. The authors avoids this pitfall by carefully re-designing the deterministic part of the core. For instance, they separate out the evaluation of the detector from the LSTM used for the core. Also, they use a fixed routine for parsing the applier instruction. The authors describe two ways of training their variant of the neural programmer-interpreter. The first is similar to the existing methods, and trains the variant using traces. The second is different and trains the variant using just input-output pairs but under carefully designed curriculum. The authors experimentally show that their approach leads to a more stable core of the neural programmer-interpreter that is close to being universal, in the sense that the core knows how to interpret commands.

I found the new architecture of the neural programmer-interpreter very interesting. It is carefully crafted so as to support expressive combinators without making the learning more difficult. I can't quite judge how strong their experimental evaluations are, but I think that learning a neural programmer-interpreter from just input-output pairs using RL techniques is new and worth being pursued further. I am generally positive about accepting this paper to ICLR'18.

I have three complaints, though. First, the paper uses 14 pages well over 8 pages, the recommended limit. Second, it has many typos. Third, the authors claim universality of the approach. When I read this claim, I expected a theorem initially but later I realized that the claim was mostly about informal understanding and got disappointed slightly. I hope that the authors consider these complaints when they revise the paper.

* abstract, p1: is is universal -> is universal
* p2: may still intractable to provable -> may still be intractable to prove
* p2: import abstraction -> important abstraction
* p2: a_(t+1)are -> a_(t+1) are
* p2: Algorithm 1 The -> Algorithm 1. The
* Algorithm1, p3: f_lstm(c,p,h) -> f_lstm(s,p,h)
* p3: learn to interpreting -> learn to interpret
* p3: it it common -> it is common
* p3: The two program share -> The two programs share
* p3: that server as -> that serve as
* p3: be interpret by -> be interpreted by
* p3: (le 9 in our -> (<= 9 in our
* Figure 1, p4: the type of linrec is wrong.
* p6: f_d et -> f_det
* p8: it+1 -> i_(t+1)
* p8: detector. the -> detector. The
* p9: As I mentioned, I suggest you to make clear that the claim about universality is mostly based on intuition, not on theorem.
* p9: to to -> to
* p10: the the set -> the set
* p11: What are DETs?

---

> ### Author Response · Authors · 2017-12-15
> **Response to Reviewer 2, Part 1**
>
> Thanks for your very constructive feedback. We have uploaded a revision to incorporate your suggestions. We will try to answer your questions and concerns one by one below.
> > First, the paper uses 14 pages well over 8 pages, the recommended limit.
> Re: We have shortened the paper from 14 to 12 pages (+refs and appendix, from 19 to 17 pages) while preserving most of the important contents by:
> 1) abbreviating the description of NPI and removing NPI's inference algorithm (Algorithm 1 in the original version) in Section 2.1;
> 2) rewriting some paragraphs (especially those in Section 5) to make them more succinct;
> 3) substantial re-typesetting (e.g. placing some figures and tables side by side, which is also common practice in other submissions). In order to present the somewhat intricate idea as clear as possible, we use in this paper quite a few figures and tables. The bad typesetting of them in the original version made the manuscript unnecessarily long.
> We'd like to mention that the 12-page revision is in fact shorter than a number of other submissions, e.g.:
> 1. Modular Continual Learning in a Unified Visual Environment (https://openreview.net/forum?id=rkPLzgZAZ), 14 pages
> 2. Towards Synthesizing Complex Programs From Input-Output Examples (https://openreview.net/forum?id=Skp1ESxRZ), 16 pages
> 3. Sobolev GAN (https://openreview.net/forum?id=SJA7xfb0b), 15 pages
> 4. N2N learning: Network to Network Compression via Policy Gradient Reinforcement Learning (https://openreview.net/forum?id=B1hcZZ-AW), 13 pages
>
> > Second, it has many typos.
> Re: We have corrected these and some other typos in the revision. We apologize for the carelessness leading to so many typos and thank you very much for the effort of pointing them out.
> * Figure 1, p4: the type of linrec is wrong.
> Do you mean that linrec has fewer arguments than shown in Figure 2? The pseudo-code in Figure 1 is only for illustration purpose. We deliberately use a simpler version of linrec to make its connection with ADD and BSTEP more apparent.
> * p11: What are DETs?
> DETs stand for detectors. The abbreviation is defined in paragraph 1 of Section 3.1.

---

> ### Author Response · Authors · 2017-12-15
> **Response to Reviewer 2, Part 2**
>
> > Third, the authors claim universality of the approach. When I read this claim, I expected a theorem initially but later I realized that the claim was mostly about informal understanding and got disappointed slightly.
> Re: The original version did lack a theorem, which is a major drawback regarding the completeness of the paper. In the revision we state the universality of CNPI with the following theorem and proposition in Section 4 (In fact we already mentioned the proposition in the original version. In the revision we propose it more explicitly):
> --------
> Theorem 1. If 1) the core along with the program embeddings of the set of four combinators and the built-in combinator _mapself are trained and verified before being fixed, and 2) the detectors for a new task are trained and verified, then CNPI can 1) interpret the combinatory programs of the new task correctly with perfect generalization (i.e. with any input complexity) by adding appliers to the program memory, and 2) maintain correct interpretation of already learned programs.
> Proposition 1. Any recursive program is combinatorizable, i.e., can be converted to a combinatory equivalent.
> --------
> Theorem 1 states that CNPI is universal with respect to the set of all combinatorizable programs and that appliers can be continually added to the program memory to solve new tasks. Proposition 1 shows that this set of programs is adequate for solving most algorithmic tasks, considering that most, if not all, algorithmic tasks have a recursive solution. We give an induction proof of Theorem 1 in Appendix C, which is newly added in the revision. The proof is in fact quite straightforward. The distinguishing feature of CNPI that enables this proof is the dynamic binding of formal detectors and callable arguments to actual programs, which makes verification of combinator's execution (by the core) and verification of their invocation (by appliers) independent of each other. In contrast, it is impossible to conduct such a proof with NPI and RNPI which lack this feature.
> For Proposition 1, instead of giving a formal proof, we propose a concrete algorithm for combinatorizing any program set expressing an recursive algorithm in Appendix B. Although we believe that Proposition 1 is true (effectively, it says that the combinatory programming language is Turing-complete), we think that a formal proof of it would be too tedious to be included in this paper. The intuition behind is that during the execution of a combinatory programs, combinators and appliers call each other to form an alternating call sequence until reaching a ACT. Arbitrarily complex program structures can be expressed in this way (see the last paragraph of Section 3.1 and Figure 3 (c) and (d)). We'd like to point out that the circle formed by the mutual invocation of combinators and appliers is a very fundamental construct in the interpretation of functional program languages. It can be seen as a "neural equivalent" of the eval-apply circle that lies at the heart of a LISP evaluator. The book "Structure and Interpretation of Computer Programs (2nd edition)" has a good discussion on this (Section 4.1.1, Figure 4.1: The eval-apply cycle exposes the essence of a computer language. https://mitpress.mit.edu/sicp/full-text/book/book-Z-H-26.html#%_sec_4.1.1). The expressive power (Turing-completeness) of functional programming languages like LISP has been well recognized. Anyway, we admit that this is a weakness regarding the theoretical rigor of this paper, which could be improved by future work.
> We have rewritten the first half of Section 4 on the universality of CNPI to state our claims more clearly. We have also replaced the conclusion section with a discussion section in the revision to discuss the potential applications of CNPI.
>
> Thanks again for the reviewing effort and any additional comments and suggestions are welcome.

---

### Author Response · Authors · 2018-01-04
**Revision 2017-12-16: Summary of Changes**

In response to reviewers' comments, we have made the following changes in the revision:
1. We have shortened the paper from 14 to 12 pages (+refs and appendix, from 19 to 17 pages) while preserving most of the important contents by:
1) abbreviating the description of NPI and removing NPI's inference algorithm (Algorithm 1 in the original version) in Section 2.1;
2) rewriting some paragraphs (especially those in Section 5) to make them more succinct;
3) substantial re-typesetting (e.g. placing some figures and tables side by side, which is also common practice in other submissions).

2. We have added a theorem and a proposition on the universality of CNPI in Section 4, along with a proof of the theorem in Appendix C.

3. We have added a discussion section to discussion one potential application of CNPI besides solving algorithmic tasks, namely natural language understanding as inferring and executing programs.

4. We have corrected a number of typos.

---

### Decision · Program_Chairs · 2018-01-29
**ICLR 2018 Conference Acceptance Decision**

**Decision:**

Accept (Poster)

**Comment:**

This paper present a functional extension to NPI, allowing the learning of simpler, more expressive programs.

Although the conference does not put explicit bounds on the length of papers, the authors pushed their luck with their initial submission (a body of 14 pages). It is clear, from the discussion and the reviews, however, that the authors have sought to substantially reduce the length of their paper while improving its clarity.

Reviewers found the method and experiments interesting, and two out of three heartily recommend it for acceptance to ICLR. I am forced to discount the score of the third reviewer, which does not align with the content of their review. I had discussed the issue of length with them, and am disappointed that they chose not to adjust their score to reflect their assessment of the paper, but rather their displeasure at the length of the paper (which, as stated above, does push the boundary a little).

Overall, I recommend accepting this paper, but warn the authors that this is a generous decision, heavily motivated by my appreciation for the work, and that they should be careful not to try such stunts in future conference in order to preserve the fairness of the submission process.